# Mapping the Most Susceptible Regions to Fire in Portugal

Tiago Ermitão [1,2,*] , Patrícia Páscoa [1,2,3] , Isabel Trigo [1,2] , Catarina Alonso [1,4] and Célia Gouveia [1,2]

1  Instituto Português do Mar e da Atmosfera, Rua C do Aeroporto, 1749-077 Lisbon, Portugal; patricia.ramos@ipma.pt (P.P.); isabel.trigo@ipma.pt (I.T.); catarina.alonso@ipma.pt (C.A.); celia.gouveia@ipma.pt (C.G.)
2  Instituto Dom Luiz, Faculdade de Ciências, Universidade de Lisboa, 1749-016 Lisbon, Portugal
3  Environmental Physics Laboratory (EPhysLab), Centro de Investigación Mariña, Campus As Lagoas, Universidad de Vigo, 32004 Ourense, Spain
4  Centre for the Research and Technology of Agroenvironmental and Biological Sciences (CITAB), Universidade de Trás-os-Montes e Alto Douro, 5000-801 Vila Real, Portugal
*  Correspondence: tmrsilva@fc.ul.pt

**Abstract:** Mediterranean European countries, including Portugal, are considered fire-prone regions, being affected by fire events every summer. Nonetheless, Portugal has been recording large burned areas over the last 20 years, which are not only strongly associated with hot and dry conditions, but also with high fuel availability in the ecosystems. Due to recent catastrophic fire seasons, Portugal has been implementing preventive policies during the pre-fire season, which, in turn, can optimize combat strategies during the fire season. In this context, our study contributes to fire prevention by identifying the regions with the highest potential to burn. The application of a Principal Component Analysis (PCA) to a range of climatological, ecological, and biophysical variables, either provided by remote sensing or reanalysis products, and known to be linked with diverse fire-vulnerability factors, allows the objective identification of the regions with the highest susceptibility to burn. The central and southernmost areas of Portugal present a stronger signal in the PCA, suggesting a likely high exposure to future fire events. The fuel accumulation over several months, in conjunction with elevation and fire weather conditions, are the terms out of the retained PCs that can explain most of the variability. The quality assessment performed for the burned areas in 2022 showed that they occurred in highly susceptible areas, highlighting the usefulness of the proposed methodology.

**Keywords:** fire susceptibility; fuel management; PCAs; fuel accumulation; vegetation productivity; land cover

## 1. Introduction

Over the last decades, an increasing trend in larger and more severe wildfires has been generally observed over many fire-prone regions, namely in the Mediterranean basin [1–3], California [4–6], and Australia [7,8]. More frequent hot and dry summer conditions, combined with climate change and high fuel accumulation over time and space, have already been shown to promote large and intense fire seasons across these regions [9–11]. Severe wildfires result in biodiversity destruction, forest damage, timber losses, carbon and nutrient cycling disruption [12–14], economic losses and human casualties, and potential post-fire effects, such as soil erosion and debris flows [15].

Portugal has a Mediterranean climate, which is predominantly characterized by hot and dry summers [16]. In this season, due to the geographical position of the country, the advection of dry and warm air from Continental Europe and North Africa can amplify the exposure of Portugal to high temperatures [17] and, consequently, to fire incidence, as 97% of extreme wildfires occurred under heatwave conditions in the period 1981–2010 [18]. Among the Mediterranean countries, Portugal has been recording a high extent of burned areas during the last decades [10]. Hundreds of thousands of hectares have burned, and a wide range of studies have shown that the recent occurrences of extreme fire seasons, such

as in 2003, 2005, and 2017, were closely linked to anomalous hot and dry summers, and also to high fuel availability, promoted by favourable conditions in the pre-fire season [10,18–20].

During the last four decades, Turco et al. [21] observed a positive trend in the number of fires, as well as in the burned area in Portugal. This positive trend can be partly related to the high number of fragmented landscapes with multiple owners (public and private) that have resulted in the strong dynamics of the country's landscape [22]. Many properties have been abandoned over the last years due to population migration from rural to urban areas [23]. As a consequence, an increase in areas covered by disorganised tall shrublands and forests, and a decrease in managed agricultural and short shrublands areas, resulted in fire risk enhancement [23,24].

In the context of the occurrence of recent large fires, and especially after the catastrophic fire season of 2017, the Portuguese government and scientific institutes have been implementing strategies and policies regarding fire prevention and firefighting [25]. Many studies have focused their analysis on improving fire prediction and assessment, using machine learning techniques, e.g., [26], and on the identification of the areas that are more vulnerable to fire occurrences. The latter usually relies on meteorological variables and fire danger ratings [27], or on their combination with morphological factors, such as elevation, slope angle, population density, fire's exposition, and land cover [28–31]. These studies strongly improved the knowledge of fire regimes in Portugal and showed where the responsible authorities should act in order to mitigate the fire risk in certain regions which are highly exposed to fire. Nevertheless, the understanding regarding the role of vegetation activity in fire susceptibility is not fully clear. The fuel accumulation and availability in ecosystems constitutes a triggering factor in enhancing fire intensity and, as pointed out by Benali et al. [32], non-managed landscapes, which have the potential to gather similar conditions to those experienced during the 2017 fire season, will promote the continuity of potentially destructive wildfires.

Within the scope of fire prevention, the identification of the most susceptible areas to burning is essential, for example, for policy and prevention decision making. Aiming to map the country's fire vulnerability, and, therefore, to identify the regions which are more prone to burn if local management practices remain unchanged, we apply a Principal Component Analysis (PCA) to a multivariable set concerning factors known to be directly or indirectly linked with wildfire activity, namely: fire risk; the energy released by fires; fire weather conditions, encapsulated in a Fire Weather Index (FWI); elevation; and information related to vegetation activity and productivity, i.e., gross primary productivity (GPP) and net primary productivity (NPP). These latter variables can provide information regarding vegetation dynamics, and, therefore, regarding the fuel accumulation available to burn, GPP being frequently used as a proxy of the amount of biomass produced over space and time [33]. Finally, we compare the PCA-based fire-susceptibility map, with the burned areas of 2022 in Portugal, i.e., with scars corresponding to fire events that took place during an independent year (not considered in the PCA).

## 2. Materials and Methods

### 2.1. Data

2.1.1. Land Cover and Vegetation Data

In this study, a range of variables was used to analyse the vegetation state. The assessment of vegetation productivity relied on satellite Gross Primary Productivity (GPP, $kgC/m^2/8$ days), namely the MOD17A2 Collection 6 product [34]. Here, we used 8-day composites with a spatial resolution of 500 m, for the period 2001–2021. The annual Net Primary Productivity (NPP) from MOD17A3 Collection 6 was extracted for the same period, and with the same spatial sampling as GPP, being expressed in $kgC/m^2/year$.

The characterization of the country's elevation is based on the Copernicus Global Digital Model (DEM) datasets, namely the Copernicus GLO-90 DEM [35] with a spatial resolution of 3-arcsecond (approximately 90 m). The Copernicus DEM datasets are subjected

to terrain corrections and hydrological editing, which allows the correction of systematic random biases [36,37].

Land cover information for 2021 was downloaded from the Portuguese General-Directorate of the Territory (DGT 2021). The annual land cover map, developed by Costa et al. [38], is constituted by thirteen land cover classes and includes the most important tree species of the country, with a spatial resolution of approximately 10 m. The accuracy of the map was estimated in 81%, making this product highly suitable for our study.

2.1.2. Burned Area and Fire Related Variables

The detection of the burned areas relied on the MODIS Burned Area product, MCD64 Collection 6 [39], with a spatial resolution of 500 m; here, we considered data for the months between June and October over the 2001–2021 period. This product showed several improvements regarding the previous version, especially in small fires detection, but also in the reduction in burn-date uncertainty. Data were generated relying on daily surface reflectance dynamics so the algorithm could detect the more likely date of fire, as well as the spatial extent of the events.

Fire weather index (FWI) corresponds to an output from the Canadian Forest Fire Danger Rating System [40]. The index has shown to be an accurate measure of fire danger across different land covers on a global scale, especially over flammability-limited environments [41,42]. In fact, Viegas et al. [43] highlighted that FWI is a particularly suitable index to monitor fire danger over the Mediterranean region. Therefore, FWI is an appropriate index for our study with the aim of monitoring fire danger over the last years.

FWI is organised into six different components: three fuel moisture codes—Fine Fuel Moisture Code, Duff Moisture Code, and Drought Code—which together integrate information regarding the effect of fuel state and availability on fire danger; the weather-driven fuel moisture information is encompassed in two indices that measure the rate of fire spread (Initial Spread Index) and the available fuel for combustion (BuiltUp Index) [40]; the sixth component, named FWI, combines all the previous information, including daily meteorological variables, namely temperature and relative humidity at 2 m, wind speed at 10 m, and 24 h accumulated precipitation. FWI was extracted from the most recent reanalysis of the European Centre for Medium-Range Weather, ERA5 [44], with daily availability and a spatial resolution of 0.25°, between 1979 and 2021.

Fire Released Energy (FRE) relies on hourly Fire Radiative Power (FRP) derived from Meteosat Second Generation (MSG) observations [45] provided by Satellite Applications Facility on Land Surface Analysis (LSA-SAF). Daily values of fire released energy are then determined for the period comprising 2004 (i.e., the start of MSG observations) to 2020, through the integration of FRP hourly values over 24 h periods on a pixel-by-pixel basis [46]. To obtain the map of characteristic released energy over the 2004–2020 period, the median value of FRE was determined at pixel-level.

The fire risk map (FRM), also generated by the LSA-SAF, combines information from Numerical Weather Prediction models, vegetation classes, and historical values of MSG FRP data to accurately characterize the fire danger [47]. Here, we used all available data from the most recent version of LSA-SAF FRM product, i.e., for the period comprising 2017 to 2021. FRM values are expressed as the percentage of the occurrence of the most extreme fire danger classes (Classes 4 and 5) in the considered period.

All the variables used in this study, with the correspondent spatial and temporal coverage, are summarized in Table 1.

*2.2. Applied Methodology*

2.2.1. Pre-Processing of Data

Different variables, with different temporal and spatial characteristics, were used in this study (see Table 1). Therefore, we worked with a common regular grid of 0.005° latitude × 0.005° longitude. Elevation and FRM data were resampled using a nearest neighbour interpolation technique, whilst FRE and FWI were resampled using

a bilinear interpolation method. Regarding the land cover map, DGT 2021, data were firstly reprojected to the geographic projection (WGS 84) and then resampled to the coarser resolution (0.005°), using a majority rule interpolation technique. After this process, land cover classes were re-aggregated into ten main categories, as described in Table S1.

**Table 1.** Variables used in this study with the correspondent source, spatial resolution, as well as their temporal coverage.

| Variable | Source | Spatial Resolution | Temporal Coverage |
| --- | --- | --- | --- |
| Gross Primary Production (GPP) | MODIS | 500 m | 2001–2021 |
| Net Primary Production (NPP) | MODIS | 500 m | 2001–2021 |
| Elevation | Copernicus GLO-90 | ~90 m | - |
| Burned Areas | MODIS | 500 m | 2001–2022 |
| Fire Weather Index (FWI) | ERA5 | ~25 km | 1979–2021 |
| Fire Released Energy (FRE) | LSA-SAF | 3–4 km | 2004–2020 |
| Fire Risk Map (FRM) | LSA-SAF | 3–4 km | 2017–2021 |
| Land Cover | DGT 2021 | ~10 m | 2021 |

The fire disturbance signature on vegetation is clearly detected on satellite products, leaving a clear signature in GPP datasets [48]. Moreover, such signature prevails during several months until vegetation recovers the normal productivity values. In order to mitigate the effect of extreme post-fire values, and considering the relatively short period of data, we excluded the value of the fire event month, as well as the 12 subsequent months, from GPP data. This procedure is performed for all burned pixels, including those that burned more than once, allowing the estimation of reference (fire-free) conditions against which we can assess the impact of fire.

2.2.2. Identification of Susceptible Areas

Fire frequency (FF) was studied by calculating the number of times that pixels burned between 2001 and 2021. The number of months elapsed since the last fire occurrence (time within burns, TwB) was also studied, allowing the observation of the fuel accumulation in ecosystems after the fire events. In this context, vegetation activity was assessed taking into account the GPP and NPP evolution. Annual anomalies of GPP ($GPP_{ANOM}$), at pixel-level, were computed through the sum of monthly anomalies that were determined in the first place, by removing the climatological median of the corresponding month. The accumulation of $GPP_{ANOM}$ was also determined for the period of 2001–2021 ($\Sigma GPP_{ANOM}$). Moreover, linear trends of GPP and NPP ($GPP_{TREND}$ and $NPP_{TREND}$, respectively) were calculated at pixel-level, considering the 2001–2021 period.

Relying on FWI, and with the aim of estimating the trend of favourable fire weather conditions in Portugal, we determined the probability of occurrence of moderate, high, very high, and extreme danger classes over the last four decades (1980–1990, 1990–2000, 2000–2010, and 2010–2020), considering fire seasons between June and October. The danger classes were previously calibrated for the Mediterranean Europe by DaCamara et al. [49], being defined as: 'Moderate' ($24.6 < FWI < 38.3$), 'High' ($38.3 < FWI < 50.1$), 'Very High' ($50.1 < FWI < 64$), and 'Extreme' ($FWI > 64$). Furthermore, in this context, we also performed an analysis of the probability of finding 'Very High' and 'Extreme' values of FWI, to infer the regions with more persistence of favourable fire weather conditions in Portugal.

Some of the variables described above are likely to be significantly correlated, especially those with clear inter-dependencies, as the cases of FRM and FRE, or GPP and NPP. Assuming that the multivariable set characterizes each pixel exposure to fire, we performed a Principal Component Analysis (PCA) to obtain an optimal (i.e., orthogonal) and smaller number of variables, which still explains most of the original dataset's variability. This statistical method converts the potentially correlated variables into an uncorrelated set, efficiently displaying the main patterns among the multivariate data [50]. The method's

efficiency was also displayed by Alonso et al. [51], who performed a PCA to characterize the exposure, sensitivity, and adaptative capacity of crops to drought events in Portugal and, also, by Bento et al. [52], who performed a PCA to evaluate the vulnerability of Iberian forests to droughts. On the other hand, PCA can also be a useful tool in the context of fire prevention, as shown by Colonico et al. [53], where the authors applied a PCA to a large set of variables, using Italy as case study.

The PCA, performed on standardized variables, identifies a reduced number of variables capable of explaining most of the variability of our dataset and allowing the finding of the most susceptible areas to burn. All the computations were based on the Python Sklearn module. The number of principal components (PCs) to be retained considers the total explained variance by $PC_i$ versus the increase explained variance by $PC_{i+1}$. To complement the analysis, the computation of the loading factors was performed, as this metric enables the determination of the contribution of each factor to each PC (Principal Component).

The final step consists of the computation of a reconstructed PC ($PC_{REC}$). Here, we considered the corresponding eigenvalues and eigenvectors, which represent the scalar and vectorial parts of the PC, respectively. Aiming to observe the most susceptible areas to fire, the $PC_{REC}$ relies on the sum of the linear combinations of each selected PC with the correspondent eigenvalue, at pixel-level, following Equation (1):

$$PC_{rec} = \sum_{i=1}^{N} PC_i \times EigValue_i \qquad (1)$$

where N is the number of PCs that explain most of the variability selected for the analysis (the first six PCs, in the present study). We then defined five classes of susceptibility, according to the output of $PC_{REC}$, being 'Low Susceptibility', 'Moderate Susceptibility', 'High Susceptibility', 'Very High Susceptibility', and 'Extreme Susceptibility'. The classes were computed following the method of Alonso et al. [51], relying on the determination of 20th, 40th, 60th, and 80th percentiles of the $PC_{REC}$ distribution signal. It is important to stress that, for this analysis, we removed the pixels with negative signal, as these points have no susceptibility to burn.

The identification of the areas with more susceptibility to burn also calls for an assessment of the land cover distribution. Thus, we calculated the percentage of each land cover category that can be found within each susceptibility class, so that it was possible to assess the distribution of different vegetation-types from lower to higher susceptibility.

### 2.2.3. Assessment for the Burned Areas of 2022

According to the preliminary report of rural fires of 2022 in Portugal from ICNF [54], a total of 10,449 fires resulted in 110,007 hectares of burned area, which represents the fifth highest value of burned area since 2012. In this context, the fire season of 2022 constitutes a suitable case study to verify the robustness of our work's method. The quality assessment of PCA relied on the selection of the top 5 events (A1 to A5) with the largest burned area extension in 2022, that can be spatially observed in Figure S3, and the assessment of the $PC_{REC}$ signal, as well as the land cover distribution, in each burn scar. We also compared these results with the distribution of $PC_{REC}$ signal of the other fires that occurred in 2022.

## 3. Results

### 3.1. Historical Vegetation Activity and Burned Area Patterns in Portugal

Portugal can be divided in two different altitude regimes due to the Montejunto-Estrela mountain range that crosses the country diagonally. The northern part of the country is characterized by higher elevations and steep mountains, especially in the central-eastern and north-eastern sectors, while the southern half of Portugal is mostly characterized by low-relief plains (Figure 1a). Nevertheless, two medium-high elevation areas are found in the southernmost part of the country.

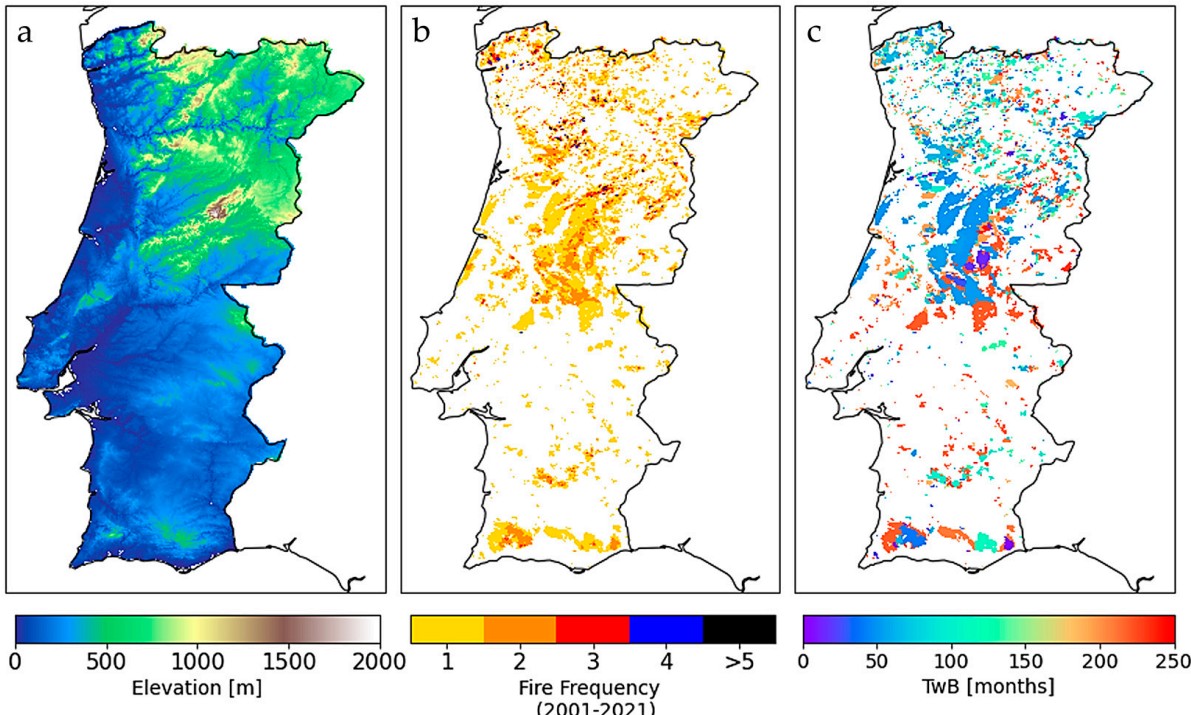

**Figure 1.** (**a**) Elevation of Portugal, in metres high; (**b**) fire frequency at pixel-level between 2001 and 2021; (**c**) time without burn (TwB) in the period 2001–2021.

The northern and central regions are the most affected by fires (Figure 1b) with a higher propensity to burn, as several areas have burned two to three times between 2001 and 2021, and a small fraction of pixels burned more than three times in this period. In southern regions, it is also possible to find areas that have burned more than once over the last 21 years.

The results in Figure 1c, that show the time elapsed (in months) since the last fire occurrence (TwB), allow the distinguishing of two main sets of large scars along the country: (i) one showing areas where previously burned vegetation remained free of fire events for more than 200 months (orange and reddish points); and (ii) another with recent fire occurrence, i.e., less than 50 months (purple and blue points). Over the areas with TwB of 100–150 months, the long-elapsed time allowed the burned vegetation to recover from the fire events and re-accumulate fuel. This is shown in Figure S1, where high accumulation of GPP, and thereby fuel accumulation, above 20 kg C/m$^2$ clearly overlaps areas with TwB > 150 months, while over recently burned areas the accumulation of GPP is about 5 kg C/m$^2$.

### 3.2. Vegetation Activity Trends and Fire Related Conditions

Sharp negative GPP$_{ANOM}$ are often found in fire scars and prevail in the following years, as in the cases of 2003, 2005, and 2017 (see Figure S2). Observing Figure 2a, $\Sigma$GPP$_{ANOM}$ presents a clear signature (negative values) over the burned areas in northern and southern parts of Portugal, being particularly pronounced in central Portugal.

The maps of $\Sigma$GPP$_{ANOM}$, on one hand, and the maps of GPP$_{TREND}$ and NPP$_{TREND}$ on the other, (Figure 2b,c, respectively) reveal contrasting patterns. A general positive GPP$_{TREND}$ and NPP$_{TREND}$ is found along the territory, with exceptions for some more recent burned areas, indicating a general increasing trend in vegetation productivity over the last two decades. The trends agree with the overall observed annual time-series, while the $\Sigma$GPP$_{ANOM}$ are strongly influenced by monthly disturbances.

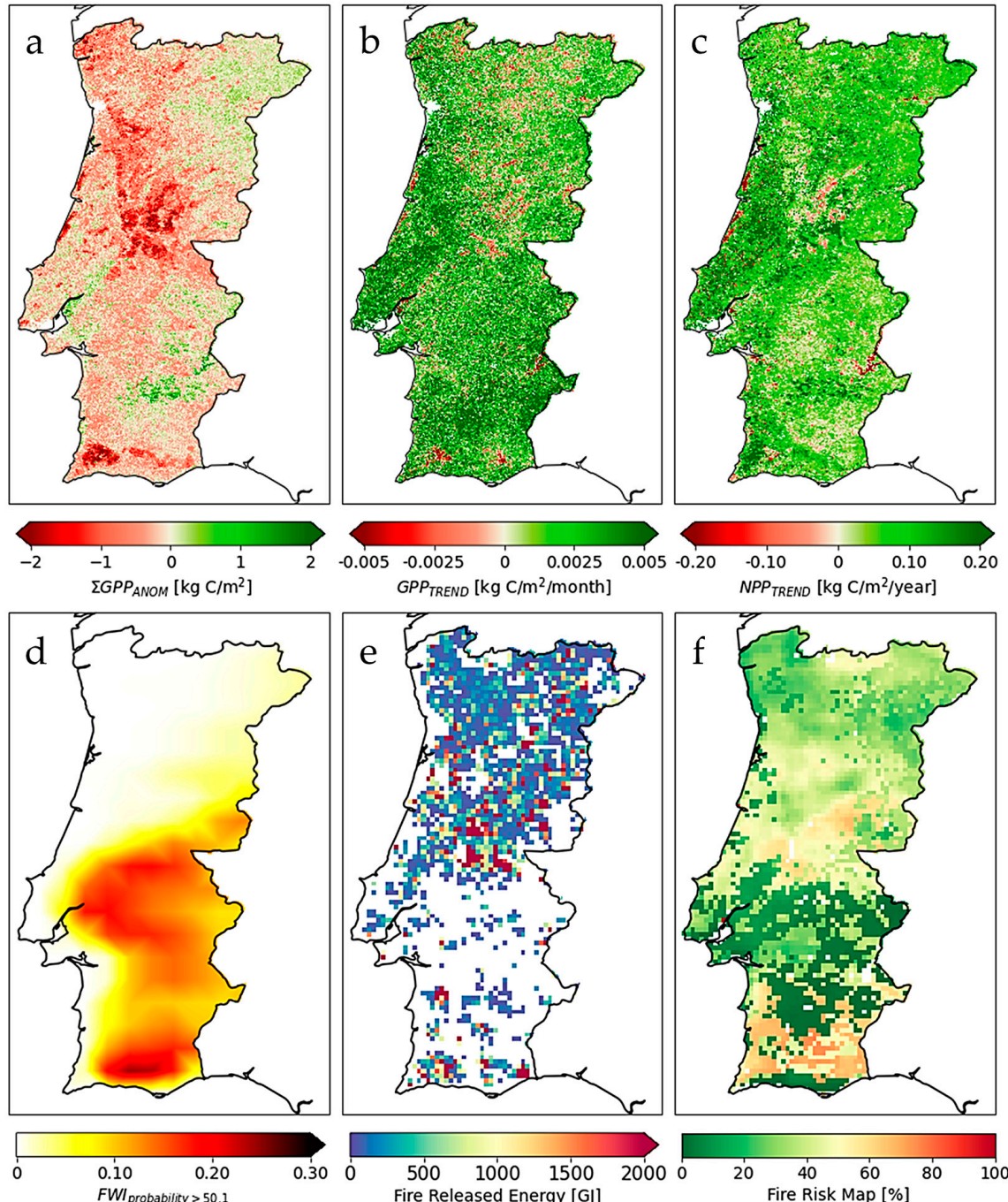

**Figure 2.** (**a**) Sum of monthly GPP$_{\text{ANOM}}$ over the period 2001–2021; (**b**) linear trend of GPP between 2001 and 2021; (**c**) same as (**b**) but for NPP; (**d**) probability of occurrence of FWI classes 'Very High' and 'Extreme' between 1979 and 2021; (**e**) median Fire Released Energy, expressed in gigajoules, between 2004 and 2020; (**f**) Fire Risk Map between 2017 and 2021. These nine variables are the ones which are used to be used in the PCA.

The areas nearer to the shore, as well as the southwest region of the country, have the highest positive trends, with GPP$_{\text{TREND}}$ of about 0.01 kg C/m$^2$/month and NPP$_{\text{TREND}}$ above 0.20 kg C/m$^2$/year.

The persistence of 'Very High' and 'Extreme' fire danger classes is noticeable in Figure 2d, where the south, as well as the centre of the country, reveal probabilities of finding these classes above 20–25%, whereas other regions have probabilities lower than 20% (as estimated for the 1980–2021 period). Besides FWI, FRE, as shown in Figure 2e,

reveals values of about 250–500 GJ over most of the points where the fire events occurred. Nonetheless, much higher values can be found, especially in the centre of the country. Intense fire occurred in these regions, promoting FRE values above 1000 GJ and, in some spots, FRE reached values even higher than 2000 GJ. In the southernmost part of Portugal, high values are also found, especially in the southwestern (about 1000–1500 GJ) and southeastern (about 1750–2000 GJ) regions. FRM, which is illustrated in Figure 2f, is higher in the centre and the south of the country, similarly to FRE. In these regions, the percentages of fire risk vary from 60 to 70% in the south and 40 to 50% in the centre of the country.

The occurrence and intensity of wildfires are closely linked to favourable weather conditions. During the last four decades, these conditions have been enhanced, as shown by FWI classes in Figure 3. A general increase in the probability of occurrence of more-severe classes of FWI has been observed throughout the territory, with a northward expansion crossing the Montejunto-Estrela mountain range (see Figure 1a). Two danger classes stand out: the 'High' and the 'Very High'. The probability of occurrence of class 'High' increased about 5% in the last four decades while the 'Very High' probability increased about 5–10% in the same period. Furthermore, the probability for the 'Extreme' class shifted from residual cases (0%) in 1980–1990 to about 3–5% in 2010–2020.

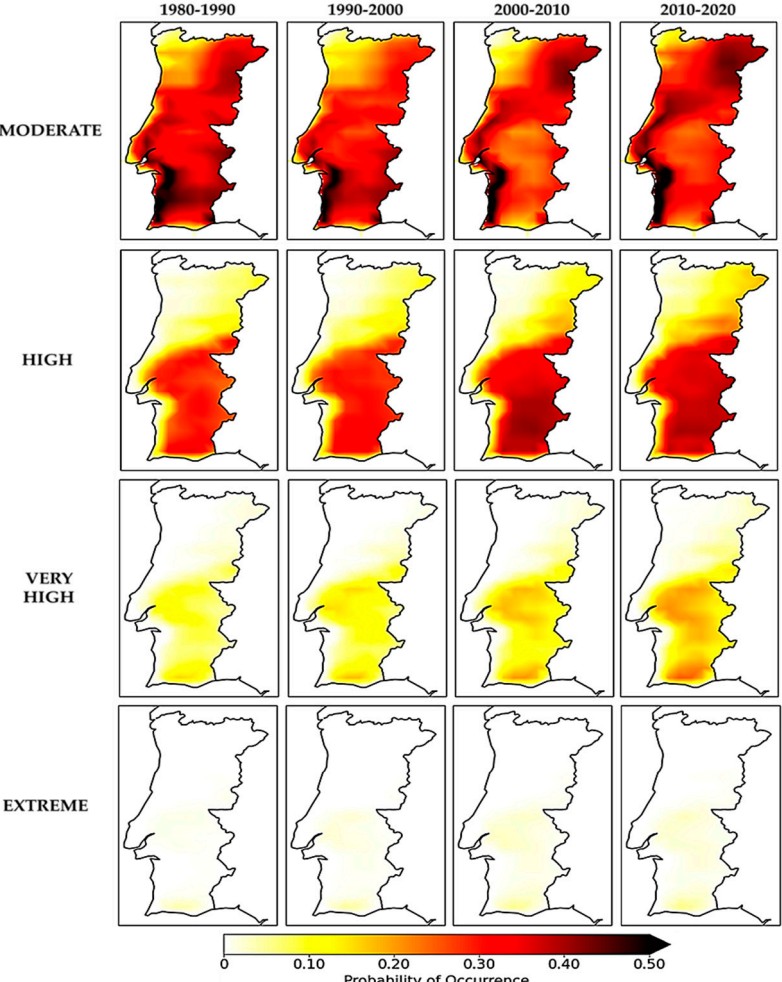

**Figure 3.** Spatial patterns of the probability of occurrence of Moderate (first line of panels), High (second line of panels), Very High (third line of panels), and Extreme (fourth line of panels) danger classes of FWI in Portugal during the last four decades.

### 3.3. Principal Component Analysis

Nine different variables, whose influences on fires were described in previous sections, were then used to perform the statistical analysis; these are summarized in Table 2.

**Table 2.** Variables considered in the PCA, aggregated into three different categories with the corresponding acronym used in the analysis.

| CATEGORY | VARIABLE | ACRONYM |
|---|---|---|
| Physical | Elevation | Elevation |
| Ecological | $GPP_{ANOM}$ accumulation along the period 2001–2021 | $\Sigma GPP_{ANOM}$ |
| | Linear Trend of GPP | $GPP_{TREND}$ |
| | Linear Trend of NPP | $NPP_{TREND}$ |
| Fire | Fire Frequency | FF |
| | Number of months elapsed since the last fire occurrence | TwB |
| | Fire Released Energy | FRE |
| | Fire Risk Map | FRM |
| | Probability of FWI > 50.1 | $FWI_{p>50.1}$ |

The explained variance of each PC, as well as the associated eigenvalues, can be observed in Table S2. The first six PCs were retained for analysis since they are responsible for almost 84% of the variance. $PC_1$ explains more than 26%, whereas $PC_2$ and $PC_3$ explain each around 14% of the variance.

The maps of the spatial distribution of each PC are illustrated in Figure 4. $PC_1$ shows a higher signal across the northern and central regions of the country and, more specifically, over points that have already burned. The loading factors (Figure 5) show that $PC_1$ has strong contributions mainly from FF and TwB. On the other hand, $\Sigma GPP_{ANOM}$ also has a strong, but negative, contribution, which is consistent with the fact that the lowest anomalies are found in burned areas.

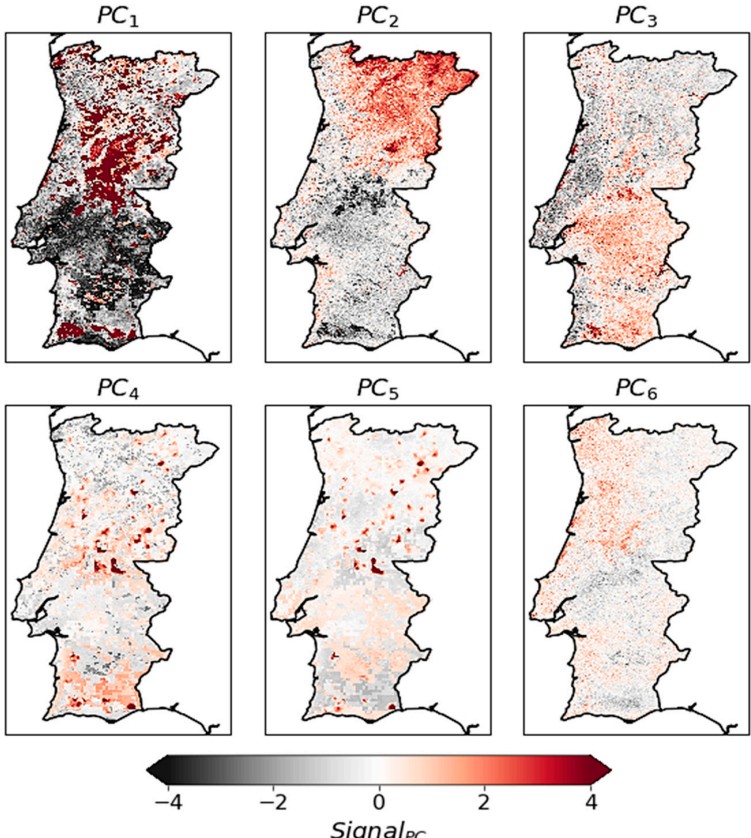

**Figure 4.** Spatial patterns of the first six PCs that explain about 84% of the total variance.

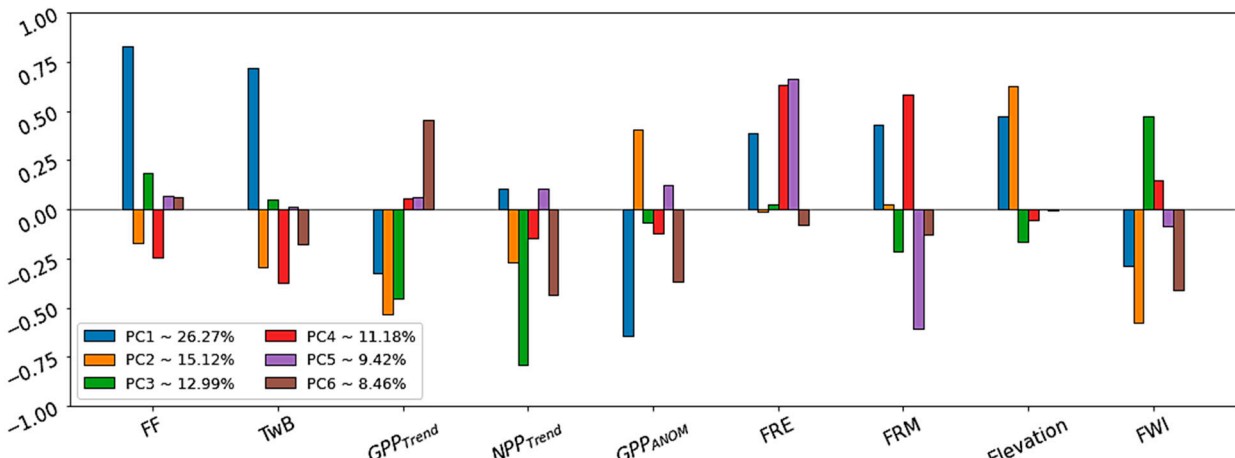

**Figure 5.** Loading factors of the chosen six PCs that reveal the contribution of each variable to each PC. The explained variance of each PC is described on the legend.

The highest signal of $PC_1$ can be found in the centre of Portugal (area with higher FF), in the north, but also in the south. The lowest signal is found between the centre and the south of the country, which is characterized by low FF.

$PC_2$ can be linked to elevation and vegetation productivity due to the strong positive signals, especially shown on the northeastern areas. In $PC_2$, the highest values are mainly located over the Iberian Peninsula Central System of mountains, whereas the weaker signal is located on the plains of the south. Additionally, in the southern region, a relevant negative contribution from $GPP_{TREND}$ and FWI is detected due to the contrasting patterns of these variables compared with the northern regions.

Regarding the $PC_3$, the stronger signal is detected in central-southern regions and, according to the loading factors, it is likely associated with the positive contribution of FWI in this area. Once again, $NPP_{TREND}$ shows a negative contribution for $PC_3$, shown by the contrast of lower vegetation productivity in this region with the higher values of the signal of $PC_3$.

The other lower-order PCs are essentially associated with FRE and FRM and, to a lesser extent, with FWI and anomalies of vegetation activity, and they capture the variance that is not accounted for by higher-order PCs.

*3.4. Land Cover Distribution of Susceptible Areas*

The maps in Figure 6 show the distribution of land cover in Portugal (left panel) and the distribution of the signal of $PC_{REC}$.

The map resulted from Equation (1) (with N = 6) and clarifies the regions where the fire susceptibility is higher, based on the weight of each PC. The areas with stronger signal (reddish spots) are mostly found in the centre and northern regions of the country whereas the southern regions are mainly characterized by negative $PC_{REC}$ signal (greyish spots). The amplification (reduction) of the signal can be strongly modulated by the landcover types of the region. Comparing the land cover distribution with the $PC_{REC}$ signal, upon first sight, the areas with high susceptibility are mainly covered by different types of tree or shrubland, while the croplands dominate the areas with less fire susceptibility.

This result stresses the importance of analysing the role of different land cover types on the potential susceptibility of vegetation to fire. Observing the land cover distribution in Figure 6 (left panel), the plains from the south of Portugal are generally covered by croplands (*Agr*) and shrublands (*Shb*), whilst the centre and north of the country, including the most susceptible regions to burn, are covered by forests, such as eucalyptus (*Euc*) and pines (*Pi*).

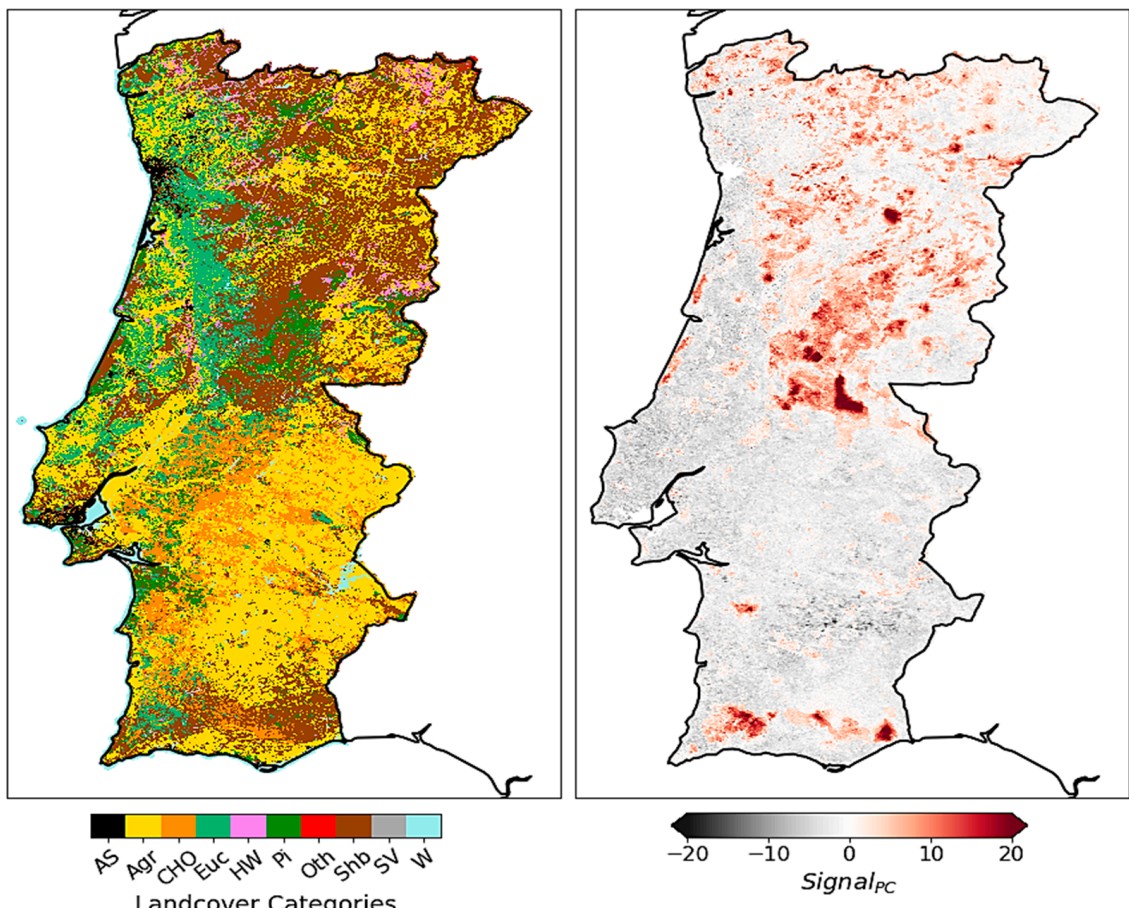

**Figure 6.** Distribution of land cover in Portugal, according to the categories previously defined (see Table S1 for classes classification) (left panel); distribution of the signal of the $PC_{REC}$.

As described before, we defined classes of fire susceptibility, considering different percentiles of $PC_{REC}$, and the values are shown in Table 3.

**Table 3.** Values of $PC_{REC}$ percentiles that defined each susceptibility class. The 95th percentile is only for observation of values within the 'Extreme' class.

|  | **p20** | **p40** | **p60** | **p80** | **p95** |
|---|---|---|---|---|---|
| $PC_{REC}$ signal | 0.94 | 2.37 | 4.40 | 7.12 | 11.71 |

The land cover distribution (see Table S1) per susceptibility class is illustrated in Figure 7. For 'Low' and 'Moderate' classes, the distribution of croplands (*Agr*), forests (*CHO*, *Euc*, *HW*, *Pi* and *Oth*), and shrublands (*Shb* and *SV*) is quite similar, varying between 25% and 40%. However, for the 'High' and higher severity classes, the distribution of land-cover types becomes uneven. The frequency of Croplands (*Agr*) decreases in higher susceptibility classes, as only about 12% (7%) of *Agr* is classified as 'Very High' ('Extreme'). In the opposite direction, an increasing percentage of shrubland (or forest) in the most susceptible classes is observed, as about 70% of the pixels classified as 'Extreme' are covered by shrublands and about 20% are covered by forest. Furthermore, an increase in the distribution of *Euc* occurs in higher susceptibility classes ('Low': 8% and 'Extreme': 13%), as well as on the *Pi* distribution from 'Low' (5%) to 'High' (8%). On the other hand, a decrease in the distribution of '*CHO*' and '*HW*' occurs in the highest susceptibility classes.

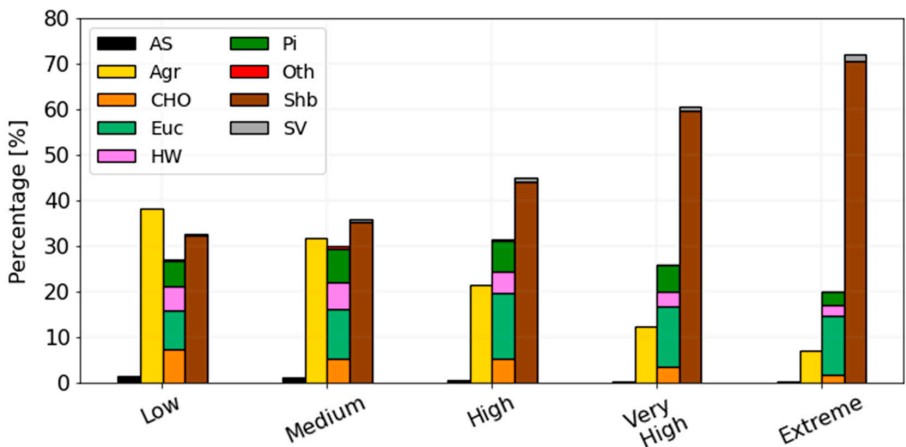

**Figure 7.** Distribution of the land cover for each class of susceptibility.

### 3.5. Susceptibility of 2022 Burned Areas

The $PC_{REC}$ signal distribution, as well as the land cover distribution, over the top 5 largest burned areas of 2022 in Portugal (see Figure S3) are represented in Figure 8. The A1 fire occurred in the National Park of Serra da Estrela, where the highest mountain of the mainland country is located, and it was the largest fire of 2022, burning around 25,000 ha [55].

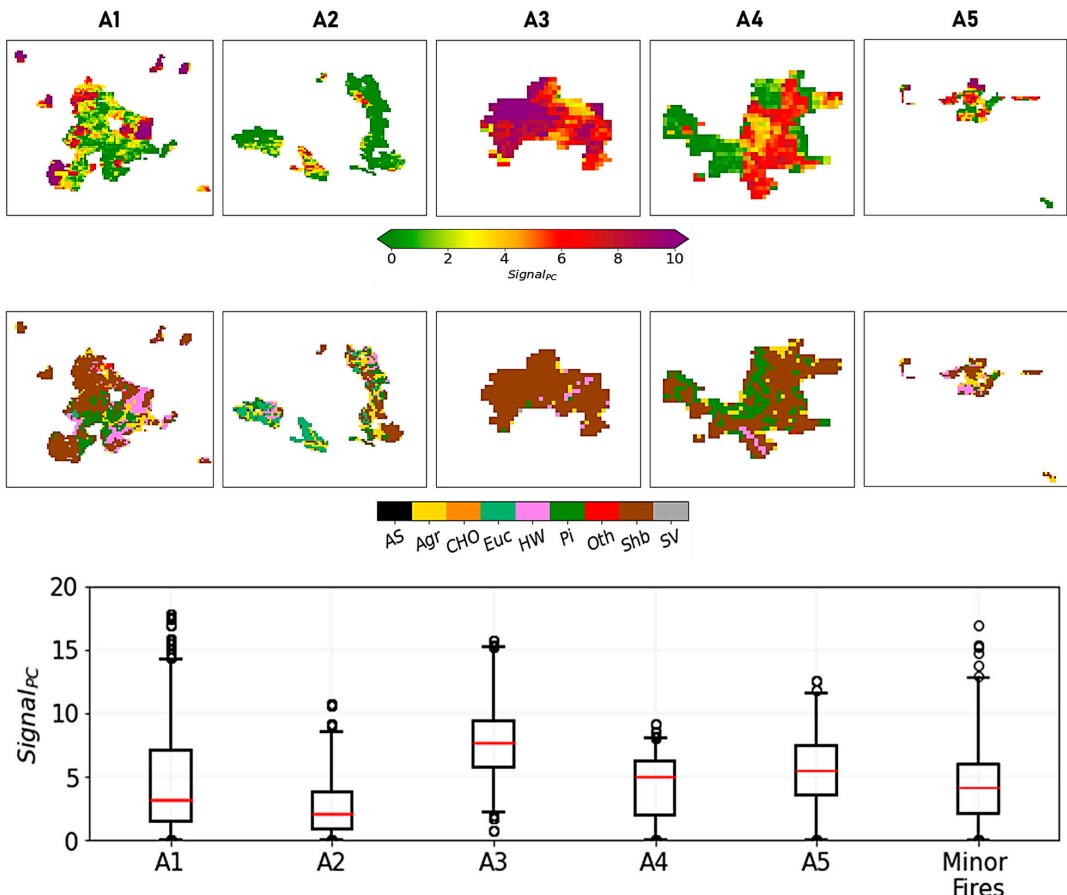

**Figure 8.** Spatial distribution of the $PC_{REC}$ (**top panels**) and land cover distribution (**middle panels**) over the top 5 largest burned areas of the fire season of 2022 (A1–A5); boxplots of the distribution of the $PC_{REC}$ signal over these areas and also over the rest of burned areas of 2022 (**bottom panels**). The whiskers of the boxplots represent the 1st and 99th percentiles. The circles represent the outliers.

According to the results, a large extension of A1 revealed high fire susceptibility, especially in the southern part of the fire scar (Figure 8, top panels). The corresponding box plot shows a large range in $PC_{REC}$ values; despite a median value of $PC_{REC}{\sim}3$, the 99th percentile (top whisker) reaches about $PC_{REC}{\sim}15$, as there are also outliers above this value, which indicate the presence of extreme susceptibility at several points. The map of the landcover shows that the burned area of A1 is mainly covered by shrublands and forests, composed of *Pi* and *HW*.

The A3 results clearly show how the area was susceptible to burn. Mainly covered by shrublands, a large part of the north-western sector of the fire scar presents a $PC_{REC}$ signal above 10, and the other sectors had $PC_{REC}$ values falling in 'High' and 'Very High' susceptibility classes. Moreover, the median value falls in the 'Extreme' class, being $PC_{REC}{\sim}7.5$.

The signal of the $PC_{REC}$ is quite strong in A4 and A5, mainly falling in 'High' and 'Very High'. Regarding the particular case of A2, despite the large burned area, there were many points classified as 'Low' and 'Moderate'.

Concerning the fires out of the top 5, the boxplots show that the distribution of the signal of $PC_{REC}$ in these areas is similar to the five major events, as the mean value is about 4.5, which falls in the 'High' category, and the highest percentiles, as well as the outliers, can be found in 'Extreme'.

## 4. Discussion

Portugal is one of the Mediterranean countries with higher burned area extension [10]. Fuel accumulation, steep mountains, and fragmented landscapes due to abandoned rural properties, in association with increasing prolonged droughts and hot conditions such as those in 2003, 2005 or 2017 [55–57], are key factors for the enhancement of fire risk in many regions. Therefore, an assessment of the critical areas that show a greater susceptibility to burning is an essential task.

Abatzoglou et al. [58] identified a pronounced likelihood of increased fire weather conditions with 2 °C global warming over the Mediterranean region as a whole, and Portugal, in particular, has already been experiencing hotter and drier conditions during summer [59]. In fact, the results of our work showed a general increase in the probability of occurrence of the most severe classes of FWI during the last years. Therefore, more intense and frequent extreme hot and dry conditions, linked to enhanced positive vegetation productivity trends (GPP/NPP and NDVI) in the Iberian Peninsula [60,61], may promote more fire events that have the potential to become more severe. In this context, a more extensive analysis regarding the wildfires' characteristics in Portugal over the last years, namely fire frequency, fire energy, and fire risk, but also elevation, fuel accumulation in ecosystems, and vegetation activity, is called for.

Pausas and Fernández-Muñoz [62] described the transition of fire regime changes in the western Mediterranean, from the twentieth century to the present day, as fuel-limited to hot and dry driven, by pointing out that climate conditions have been exacerbating their role in wildfires amplification, surpassing the influence of land use changes in these events. On the other hand, Mateus and Fernandes [63] and Silva et al. [22] highlighted that the fire activity in Portugal is strongly modulated by land use conflicts and fragmented landscapes with a prevalence of highly unmanaged flammable vegetation. In our work, the results of the PC showed a strong signal over regions with high fuel accumulation over several months, which highlight that, in Portugal, fuel management continues to be a key factor for fire propagation, flammability, and intensity. Therefore, although we do not directly address an analysis for hot and dry conditions in ecosystems, our results of fire weather conditions are in line with Pausas and Fernández-Muñoz's [62] insights. Moreover, the results also agree with Mateus and Fernandes' [63] observations, in the way that the extreme fire seasons in Portugal are strongly linked to the association of fuel accumulation availability with hot and dry conditions, which was also pinpointed by Gouveia et al. [18] and Ermitão et al. [19].

Some of the most susceptible areas to fire have burned in the last years, especially in 2017, when wildfires devastated the central part of the country [10,19,20]. Our analysis relied on a dataset of around 20 years (depending on the variable) and indicated that, despite the recent occurrence of fire events, those areas constitute a hotspot and have a great potential to burn in the near future. Moreover, the results also show that the most extreme classes of fire susceptibility were found over areas mainly covered by shrublands and forest, such as eucalyptus, or a mixture of both. Considering that shrublands and eucalyptus are vegetation types very prone to wildfire spread, this can constitute an enhancement of the fire risk over these areas.

The quality assessment of the PCA was performed for the fire season of 2022 in Portugal. The analysis of the $PC_{REC}$ signal distribution across the top 5 burned areas allowed the observation that fire susceptibility was mainly high to extreme. The A1 event revealed several spots classified as extremely susceptible to fire. Moreover, the south border of the fire scar, where the fire event is likely to have started [54], was extremely susceptible to burn, which constitutes an important factor to promote the amplification of the fire's intensity and severity. In the case of A2, despite low to moderate values of susceptibility, the presence of some points in the south and north-eastern part of the fire scar which were highly susceptible to fire, in conjunction with potential vegetation stress caused by hot and dry summer conditions [54], could be the triggering factor for this event. This analysis also enabled the finding of high $PC_{REC}$ signals in other minor fires, indicating that, despite the favourable conditions of vegetation for burning, the events were possibly mitigated earlier due to external factors, e.g., firefighters, efficient combat, unfavourable fire weather conditions.

Studies related to the identification of areas with wildfire susceptibility in Portugal have been performed during the last years, mainly since the harsh fire season of 2017. The centre of the country was strongly disturbed by the wildfires of this year [10,19] and these events left deep scars in vegetation, which remained for a long period. Relying on variables related to elevation and slope angle, which have a strong influence on vegetation distribution, flammability and local climate [64], and land cover and fire weather indices, these studies mapped the areas with high exposure and vulnerability to fire in the country [23,29,31]. In our work, besides those variables, the applied method also addresses variables related to vegetation activity and fuel accumulation. The results not only agree with the findings of these studies, but also provide new insights regarding the role of vegetation activity, which was revealed to be a major driver and contributor in exacerbating the ecosystems' susceptibility to burn. The study also contributed to fire prevention in Portugal, allowing national authorities to apply efficient policies over the areas that are more susceptible to burn. The approach used here allows the assessment of the areas that need special attention, especially regarding biomass control and landscape management. Furthermore, although we do not address the meteorological variables that have an important role in mediating fire intensity (such as air temperature, wind speed, soil moisture or relative humidity), the considered variables can be updated periodically (e.g., yearly or bi-yearly), revisiting the identification of fire susceptible regions before the start of each fire season.

In the context of climate change, with hotter and drier summers, the ecosystems' susceptibility, as well as the exposure and human vulnerability to fire events, will increase [30]. Nevertheless, fuel management may constitute a key factor in partially mitigating the occurrence of increasingly large and intense wildfires in Portugal and may avoid the return of catastrophic fire seasons as the ones experienced in Portugal in 2003, 2005, and especially 2017.

## 5. Conclusions

Favourable fire weather conditions have been increasing during the last decades over the Mediterranean region, and thereby in Portugal, due to climate change. In association with accumulated and unmanaged fuel in many areas, large fires have occurred in the

last 20 years in Portugal, leading to severe economic, social, human, and ecological consequences. In this context, this study aims to identify and characterize the regions with the highest susceptibility to burn in the country, considering a multivariable set of variables related to fires.

The PCA performed here showed that the availability of fuel in ecosystems plays an important role in mediating the vegetation's susceptibility to fire, allowing the inference that the continuity of unmanaged landscapes, especially in the northern and centre regions of Portugal, will promote more flammable and intense wildfires in the next years. In fact, the most highly fire-susceptible areas are mainly covered by shrublands or forests, or a mixture of both, which constitutes an enhancement of fire risk in many areas.

The assessment of the applied method relied on the study of the fire season of 2022, allowing the conclusion that this set of variables is suitable for the purpose of the study and to show that the presence of hotspots of vegetation's susceptibility in some regions can promote large fires, as in the case of A1. The overall results also show that fuel management policies are essential in reducing, to a certain extent, the impact of wildfires in fire-prone hotspots like Portugal. Moreover, considering that hotter and drier conditions in summer will amplify the fire risk across wide regions in the coming years, the effectiveness of land management can be a strong way to partly mitigate the impact of fires.

**Supplementary Materials:** The following supporting information can be downloaded at: https://www.mdpi.com/article/10.3390/fire6070254/s1, Figure S1: GPP accumulation in burned areas since the last fire event occurrence; Figure S2: GPP$_{ANOM}$ in Portugal, between 2001 and 2021; Figure S3: Map of burned areas of the fire season of 2022 in Portugal for the months of June (yellow spots), July (orange spots) and August (red spots); Table S1: Categories and sub-categories of land cover classes used in this study. The acronym, as well as the representative color used, are identified; Table S2: Explained variance of the first six PCs and the total amount of variance explained by these PCs. The corresponding eigenvalues are also expressed.

**Author Contributions:** All authors participated in the conceptual design of the study and contributed to the interpretation and analysis of the results, as well as the redaction of the manuscript. T.E. and P.P. made the calculations, figures and tables. T.E. wrote the manuscript. P.P. and C.G. designed the methodology and defined the datasets. I.T. contributed to improving the analysis and redaction of the manuscript. C.A. provided the calculations of fire variables and contributed to the redaction of the manuscript. Each of the co-authors performed a thorough revision of the document, providing useful advice on the intellectual content. All authors have read and agreed to the published version of the manuscript.

**Funding:** This study was supported by the FCT (Fundação para a Ciência e a Tecnologia, Portugal) through national funds (PIDAAC)—UIBD/50019/2020, under the project FlorestaLimpa (PCIF/MOG/0161/2019). The study was also performed within the framework of LSA-SAF, co-funded by EUMETSAT, and it was also partially supported by the European Union's Horizon 2020 research project FirEUrisk, with the Grant Agreement no. 101003890.

**Institutional Review Board Statement:** Not applicable.

**Informed Consent Statement:** Not applicable.

**Data Availability Statement:** No new data were presented in this study. Reanalysis and remote sensing data are public.

**Conflicts of Interest:** The authors declare no conflict of interest.

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
