# Peer review of "Mapping the Most Susceptible Regions to Fire in Portugal"

_fire, doi:10.3390/fire6070254_

Round 1

Reviewer 1 Report

In the context of climate change, many regions around the world are facing the impact of fires. How to use quantitative research methods such as remote sensing and GIS to identify areas prone to fire and make necessary preparations in advance is of great practical significance for the development of various regions. 

This article provides a good analysis and research on the fire risk in Portugal, providing many useful suggestions for future fire prevention work in this region.  I propose the following suggestions for the authors' reference.

1- The research background section can explain some basic climatic and environmental characteristics of Portugal, especially the Mediterranean climate in the same region as Portugal. These climate characteristics are of great significance for the fire prevention work in this natural geographical area.

2- Will the spatial resolution of different thematic maps in Figure 1 be inconsistent, which will have an impact on the research results? This issue can be appropriately explained and supplemented.

3- PCA method is a relatively mature method and research means in Spatial analysis and environmental remote sensing. The author can properly explain the research background of this method in fire prevention and other fields.

This article has good readability, smooth grammar and language.

Reviewer 2 Report

General Comments:

The manuscript entitled “Mapping the Most Susceptible Regions to Fire in Portugal” aims to help and contribute to policy and fire prevention decisions by identifying the areas with the highest probability to burn. Portugal, the study area, is a fire-prone Mediterranean area, and recently faced the largest burned areas of the last two decades. For this purpose, the authors identified and mapped the most susceptible areas to fire through the application of a PCA- Principal Component Analysis, by using a comprehensive set of variables (climatological, ecological, biophysical), gathered from remote sensing and reanalysis data. The authors applied a PCA on the following variables: fire risk - the energy released by fires; fire weather conditions, FWI - elevation - information on vegetation activity and productivity. Finally, the outcomes of the PCA-based maps were compared to the burned areas of 2022.

The manuscript is overall interesting and properly-written. The introduction section gives a comprehensive state of the art and, in my opinion, is good. The methodology in M&M is clear; the results are detailed, and the discussion is well-written. However, there are some points that should be better addressed in order to improve the clarity and the quality of the manuscript. The main weaknesses of the work are summarized below, while other comments are provided in the Specific comments.

Overall, the main weaknesses of the work are in the results section. Here, the authors described and discussed their outputs, but comparison with other works should be placed in the discussion section. Overall, the first part of the Results is disorganized, needs to be written more clearly and orderly; some figures and tables do not follow the text, and some of them are poorly described (e.g. Fig.3). Figures distant from their description make the reading tiring. In general, the results section needs to be improved and better organized. Overall, maps can be enhanced (also in the Supplementary material), for example adding a digital elevation model and/or other similar layers (e.g. hillshade or aspect). For these reasons, I suggest the publication of this work in the Fire journal after some minor revisions will be performed.

Specific Comments:

L33-35 please evaluate a phrase like this - More frequent hot and dry summer conditions, combined with climate change and high fuel accumulation over time and space, have already been shown to promote large and intense fire seasons across these regions.

L46 please consider an easier expression like “During the last four decades, Turco et al. [18] observed a positive trend in the number of fires and the burned area in Portugal”

L57 please consider using past simple instead of present perfect continuous

L86 perhaps you can change the title paragraph to “Land cover and vegetation data”

L236 I think that Fig1a and 1b are material and methods, while the others are results of the work. In the caption, I would suggest using fewer acronyms (e.g., TwB, GPPANOM, NPP...)

L241 I would suggest the authors not use future tense forms

L245 I would suggest the authors add a few or any references in the results section. Instead, I would propose to shift them to the M&M or to the Discussion part.

L253-257 I think this description is a part of M&M section, connected to figure 1a,  not a result

L259-262 is the same as above. This description is about the fire regime, relative to the Fig1b (which is in the M&M section)

I would suggest separating thousands with a comma rather than the space

L280 In the results section, the authors are supposed to describe the results of the work and not discuss and compare them with other works

L294 If possible, I suggest adding a layer concerning elevation in the maps

L312 I propose to shift here the Fig2

L327 Please describe what PCs, PC2, PC3, and the following ones mean.

L341 Please describe better the utility of these compared maps

L368 Please add here Fig5

L443-458 Please exclude the names of the authors cited if there is a connected number to the reference.

Overall, the article is well-written. The results section should be improved in clarity and quality.
